# Preparation of Carbon-Based Solid Acid Catalysts Using Rice Straw Biomass and Their Application in Hydration of α-Pinene

**Zhaozhou Wei, Deyuan Xiong * Pengzhi Duan, Shilei Ding, Yuanlin Li, Lisi Li, Peirong Niu and Xusong Chen**

1   Guangxi Key Laboratory of Petrochemical Resource Processing and Process Intensification Technology, School of Chemistry and Chemical Engineering, Guangxi University, Nanning 530004, China; zzwei0608@outlook.com (Z.W.); pzduan0306@163.com (P.D); dshilei1234@126.com (S.D.); lylgzgy@163.com (Y.L.); minzhuli123@outlook.com (L.L.); peirongn@outlook.com (P.N.); chenxus1@outlook.com (X.C.)

*   Correspondence: dyxiong@gxu.edu.cn; Tel.: +86-1387-718-2465

**Abstract:** Carbon-based solid acid catalysts were prepared using rice straw (RS) waste, and the effects of carbonization temperature and sulfonation temperature on the catalytic activity were investigated. The properties of the catalysts were characterized using thermo gravimetric (TG), scanning electron microscope (SEM), Brunauer–Emmet–Teller (BET), Fourier transform infrared spectroscopy (FT-IR), temperature-programmed desorption (TPD), and X-ray photoelectron spectroscopy (XPS), and their activities were investigated through the hydration of α-pinene. The conversion of α-pinene and the selectivity of α-terpineol reached 67.60% and 57.07% at 80 °C and atmospheric pressure in 24 h, respectively. The high catalytic capacity of the catalyst is attributed to the high acid site density and high porosity of the catalyst. TPD analysis and FT-IR spectroscopy showed that the catalyst produced by low-temperature carbonization at 300 °C followed by low-temperature sulfonation at 80 °C had abundant strong acid sites (0.82 mmol/g), which can effectively inhibit the side reactions of hydrated α-pinene. The total acidity reached 2.87 mmol/g. $N_2$-physisorption analysis clearly indicated that the obtained catalysts were mesopore-predominant materials, and the $S_{BET}$ and $V_{Total}$ of catalysts reached 420.9 $m^2/g$ and 4.048 $cm^3/g$,, respectively. Preparation of the catalyst involves low energy consumption, and its cheap raw materials make the whole process simple, economical, and environmentally friendly.

**Keywords:** biomass; carbon-based catalyst; hydration; α-pinene; α-terpineol

## 1. Introduction

Turpentine is a valuable and renewable natural resource widely used in the medical industry [1]. It is easy to obtain by the distillation of pine sap. The major component is α-pinene, but its derivative α-terpineol has a higher value and a pleasant odor similar to that of lilacs and is therefore widely used in perfumes, cosmetics, and cleaning agents[2]. α-Terpineol is a monocyclic monoterpene tertiary alcohol and is found in plants. However, only relying on the extraction of α-terpineol from plants does not meet the daily needs of humans due to its low productivity. Therefore, the preparation of α-terpineol with α-pinene has received more attention in recent years.

At present, there are two major methods for the synthesis of α-terpineol with α-pinene: the homogeneous method and the heterogeneous method. The homogeneous method uses a liquid acid. For example, the traditional way to prepare α-terpineol is using dilute sulfuric acid to catalyze

turpentine. Prakoso et al. used chloroacetic acid as the catalyst in the hydration of $\alpha$-pinene, wherein the conversion of $\alpha$-pinene and selectivity of $\alpha$-terpineol reached 99.00% and 70.00%, respectively [3]. However, the excess of liquid acid cannot be recycled from the reaction mixture sufficiently. In addition, equipment corrosion and environmental contamination are serious problems in the traditional process of $\alpha$-terpeneol preparation.

The heterogeneous method generally uses a solid acid catalyst. Yang et al. studied the hydration of $\alpha$-pinene using a pilot-scale jet reactor and Amberlyst 15 catalysts in aqueous ispropylamine at the temperature of 70 °C. The Amberlyst 15 catalysts showed good activity (93.12% conversion with 35.20% selectivity of $\alpha$-terpineol) and the pilot-scale jet reactor increased the mass transfer between $\alpha$-pinene and catalysts [4]. Maria et al. prepared inorganic oxide-supported trichloroacetic catalysts for the hydration of $\alpha$-pinene. The conversion of $\alpha$-pinene and selectivity of $\alpha$-terpineol reached 57.00% and 75.00%, respectively. The carboxylate group of trichloroacetic acid incorporated Zr to form a six-membered ring, which facilitated the increase in the carboxylate group loaded onto the support and improved the Brönsted acidity [5]. Valente and Vital studied the hydration of $\alpha$-pinene by ultra-stable Y zeolite catalysts in aqueous acetone. The conversion of $\alpha$-pinene and the selectivity of $\alpha$-terpineol reached 90.00% and 55.00%–60.00%, respectively. The ultra-stable Y zeolite catalysts showed increased activity and selectivity to $\alpha$-terpineol because of the gradual improvement in the number of Lewis acidity sites. They also successfully carried out the preparation of $\alpha$-terpineol over composite polydimethylsiloxane membranes filled with beta zeolites or sulfonated active carbon, resulting in good selectivity (70%–75% at 90% conversion) [6,7]. These results demonstrated that solid acid catalysts were successful for the hydration of $\alpha$-pinene. Solid catalysts can be easily recycled and present lower environmental risks. However, the cost of these catalysts is high, and problems such as the requirement of expensive catalyst support and high energy consumption still exist.

Agricultural production has produced a large amount of waste biomass every year, which is normally used as a low-value energy resource, burned in the field, or even thrown away. Agricultural waste utilization is a challenging issue. Some researchers used the incomplete carbonization of agricultural waste as biomass carbon catalysts with the superiority of clear textural properties, abundant porous structures, and high catalytic activity. The wide application of biomass carbon catalyst will provide a new route for decreasing the agricultural waste issue. Biomass carbon-based catalysts have received more attention in the last 10 years. For example, corncob residual and palm empty fruit bunch have been used as the raw material for catalysts for synthetic biodiesel [8]. However, using agricultural waste as raw materials to prepare biomass carbon-based catalysts in the hydration of $\alpha$-pinene have not been extensively studied. Xie et al. [9] used kraft lignin as raw materials to prepare biomass carbon-based catalysts through incomplete carbonization (400 °C) with loaded phosphoric acid and –$SO_3H$ groups at 180 °C. The conversion of $\alpha$-pinene and the selectivity of $\alpha$-terpineol reached 95.30% and 55.30%, respectively. The pore structure of catalysts could be controlled by adjusting the phosphoric acid dosage during lignin carbonization, and the high mean pore size (6.73 nm) promoted the hydration of $\alpha$-pinene in the interior of the catalyst. However, the preparation of catalysts is too difficult, and the cost of kraft lignin purification is high. Moreover, it caused a waste of raw materials by producing a large amount of by-products.

The rice straw is gramineae species and mainly composed of cellulose. The contents of cellulose, hemicellulose, and lignin of rice straw were approximately 43.15%, 16.96%, and 16.64%, respectively [10]. There are more vascular bundles and catheters in the rice straw, which suggests the presence of a well-developed pore structure and medium mechanical strength in the rice straw. Thus, the rice straw allows more sulfuric acid to enter the pores, resulting in the increase in the number of strong acid sites.

In this study, in order to reduce the cost during the production of $\alpha$-terpineol, improve the yield, and decrease the content of by-products, we devised an easy strategy to prepare biomass carbon-based solid acid catalysts. The incomplete carbonization of waste rice straw will be used as support to prepare biomass carbon-based solid acid catalyst through low-temperature sulfonation

with concentrated H₂SO₄. Compared with kraft lignin, the energy consumption of prepare rice straw carbon-based solid acid catalysts are lower; the temperature of carbonization and sulfonation were 300 °C and 80 °C, respectively. The abundant strong acid sites (0.82 mmol/g) of carbon-based solid acid catalyst, which can effectively inhibit the side reactions of hydrated $\alpha$-pinene, lead to the yield of by-products decrease. The production process route of $\alpha$-terpineol provided in this paper develops a new way for the use of agricultural waste and the production of $\alpha$-terpineol. The development and application of this type of biomass carbon-based solid acid can effectively reduce the emissions of rice straw waste. It is helpful for the protection environment and waste utilization. The catalysts were characterized, and the effect of sulfonation and carbonization temperature on the catalytic activity was explored.

## 2. Results

### 2.1. Thermal Stability

The thermal stability of RS300 was examined by a synchronous thermal analyzer. It is shown in Figure 1. A preliminary loss of weight occurred at 30–150 °C, indicating that the absorbed water broke away from RS300 by evaporation. The slight loss of quality at 150–240 °C indicated that the partially bound water was removed. The next dissociation zone in the temperature range of 240–400 °C indicated that carbonization at higher temperatures might lead to an excessive breakdown of the macromolecules and removal of organic portions. For example, the glycosidic bonds of the cellulose and hemicellulose structure of RS300 had an open-loop fracture, leading to the formation of some new products that contained carbon–oxygen double bonds (i.e., carboxyl and carbonyl) and volatile compounds with low molecular weights (i.e., carbon dioxide and carbon monoxide). At around 400–800 °C, more loss of weight took place, and the crystallization zone of cellulose and hemicellulose of RS300 were greatly damaged, leading to a decreased polymerization degree in its structure, which may be assigned to the transformation of cellulose and hemicellulose structure into a graphite structure [11]. Thus, the carbonization temperature during catalyst preparation was selected as a low-temperature value (<400 °C) to retain the primitive structures of RS, as carbonization at higher temperatures might lead to the formation of a rigid structure and be bad for loading −SO₃H groups.

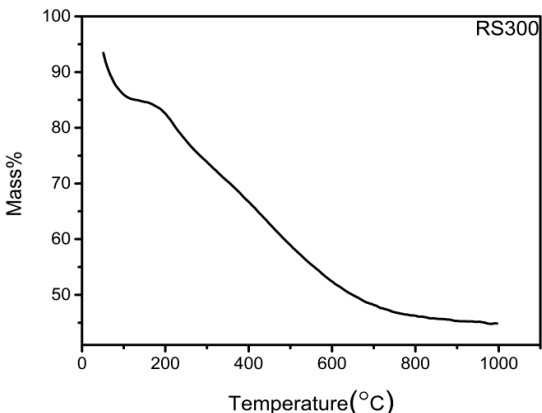

**Figure 1.** Thermo gravimetric analysis spectra of RS300.

### 2.2. Characterization of Catalysts

The morphology of carbon precursors and the corresponding carbon-based solid acid catalysts were analyzed by SEM, as shown in Figure 2. The surface of RS240 and RS300 were curly and smooth. However, RS350 showed irregular and cracked surfaces when the temperature of carbonization was increased. This phenomenon was attributed to carbonization at a lower temperature, the primitive structures of RS wasn't destroyed. As the temperature of carbonization

increased, and might lead to excessive breakdown of the macromolecules and cracking of glycosidic bonds. The morphological characteristics of RS240-80 presented massive tearing structures. It might be that the RS retained partially bound water and the structure of original cellulose in the low-temperature carbonization (e.g., 240 °C), leading to glucose reaggregation during the cellulose hydrolysis of RS240 in concentrated $H_2SO_4$. Therefore, the SEM images of RS240-80 demonstrated a collapsed and lumpy structure. As the carbonization temperature rose, the morphological structure of RS300-80 was nearly the same as that of the corresponding carbon precursor. One possible explanation for this phenomenon is that the partial breakdown of the hydrogen bonds of macromolecules and complete removal of bound water at 300 °C caused a decrease in the extent of glucose reaggregation during cellulose hydrolysis of RS300 in the concentrated $H_2SO_4$. Similarly, well-developed tubular porous features were observed in RS300 and RS300-80. However, the SEM images of RS300-80 showed an irregular rough surface compared to that of RS300. This may be attributed to the formation of a "soft" structure by low-temperature carbonization, which could have allowed more sulfonic acid ($-SO_3H$) groups being loaded on RS300 [12], resulting in the slight change on the surface of RS300 during the sulfonation process. In addition, RS300 had abundant pores with different sizes and a smooth surface, and this also provided more loading sites of sulfuric acid, which resulted in a higher covalent binding energy for carbon and sulfonic acid ($-SO_3H$) groups [13]. When the carbonization temperature was above 300 °C, RS350 became harder, and the flexibility of the polycyclic aromatic carbon decreased through plane growth and carbon sheet stacking. Thus, the general texture of RS350-80 showed little changes after sulfonation. As the temperature of carbonization increased, the pore structure of the catalyst became more abundant (<300 °C), which further increased the temperature, and the overall structure of the catalyst appeared split and faulty. The former was attributed to the "soft structure" of precursor formed under low temperature carbonization, which was easily changed by concentrated sulfuric acid in the process of sulfonation, including dehydration, hydrolysis, and $-SO_3H$ groups loading. The latter RS forms a "hard structure" under high-temperature carbonization, which was difficult to be changed by concentrated $H_2SO_4$.

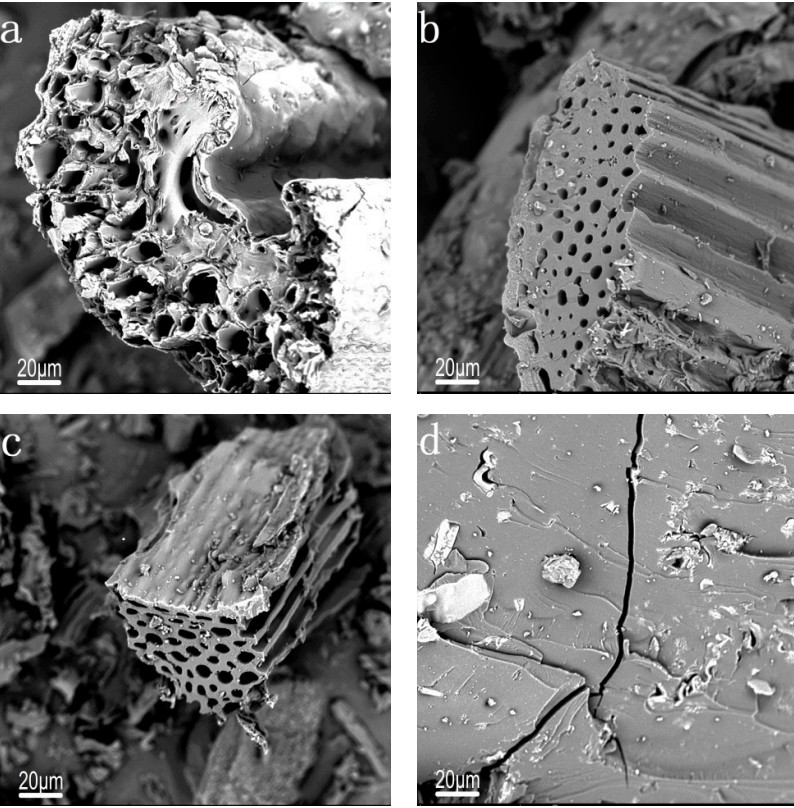

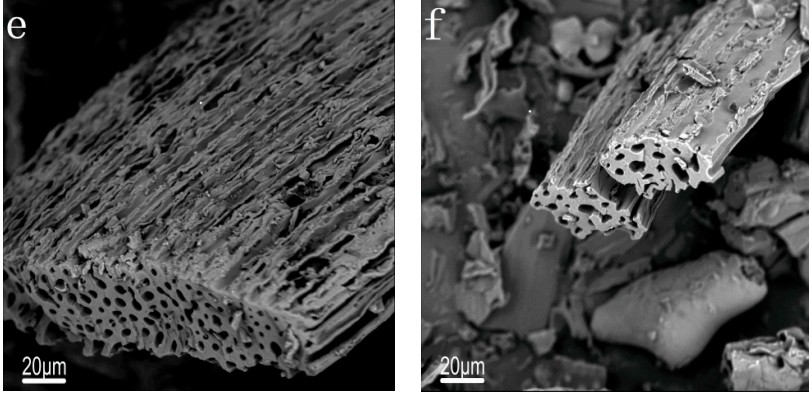

**Figure 2.** SEM images of RS240 (**a**); RS300 (**b**); RS350 (**c**); and corresponding catalysts (**d**) RS240-80; (**e**) RS300-80; and (**f**) RS350-80.

*2.3. $N_2$-physisorption*

The pore structure of solid acid catalysts significantly influences the reaction of the heterogeneous system. The $N_2$-physisorption results for the carbon precursor and catalysts are presented in Table 1. As the carbonization temperature increased, the $S_{BET}$ and $V_{Total}$ of the carbon precursor first decreased and then increased. It may be due to the presence of bound water that the surface morphology of RS240 underwent expansion and curl, leading to a high $S_{BET}$ and $V_{Total}$. As the bound water was removed at 300 °C, the $S_{BET}$ and $V_{Total}$ of RS300 decreased. While carbonization occurred at a higher temperature (>300 °C), it could have led to broken glycosidic bonds of cellulose and hemicellulose by high-temperature pyrolysis and the removal of organic portions of the carbon precursor. Thus, the $S_{BET}$ and $V_{Total}$ increased. The $S_{BET}$ and $V_{Total}$ of RS240-80 was lower than those of RS240, indicating that structural changes had occurred in the carbon structure after sulfonation, including collapse owing to the dehydration reaction in the concentrated $H_2SO_4$ and degradation of the carbon.

The $S_{BET}$ increased from 260.5 to 420.9 $m^2/g$ in RS300 after sulfonation. Furthermore, it was found that RS300-80 had a higher $V_{Total}$ (4.05 $cm^3/g$) compared to RS300, this may be a result of partial pore and surface cracking in RS300 by glycosidic bond brerakage of cellulose and hemicellulose during the sulfonation process[14]. RS300 still maintained some of the original structure of cellulose and hemicellulose after low-temperature carbonization (300 °C) but it was hydrolyzed in acidic solutions with the decrease in polymerization degree and mechanical strength, resulting in the internal channel and surface of RS300-80 cracking. Therefore, RS300-80 had larger $S_{BET}$ and $V_{Total}$. The SEM images also confirmed a similar external surface for RS300-80.

The $S_{BET}$ and $V_{Total}$ of RS350 increased slightly after sulfonation, indicating that the sulfonation temperature (under 80 °C) had little effect on the $S_{BET}$ and $V_{Total}$ of RS350-80. According to TG analysis, an increase in carbonization temperature may result in a decrease of alkyl side chains and oxygen-containing groups. Therefore, it can be inferred that the cellulose and hemicellulose structure of RS350 was damaged under high temperature carbonization, and caused a decrease in the extent of cellulose hydrolysis of RS350 in the concentrated $H_2SO_4$, so that the internal channels and surfaces of RS300-80 were not easily broken. Thus, the $S_{BET}$ and $V_{Total}$ of RS350-80 only slightly increased. With the carbonization temperature increasing, the $S_{BET}$ and $V_{Total}$ of the catalysts first increased and then decreased. This indicates that an appropriate increase in the carbonization temperature was useful for improving the porous properties. However, an excessively high carbonization temperature, e.g., 350 °C, likely causes the RS350 to become hard by RS carbonization at high temperatures, thus leading to it being difficult to be modified by concentrated sulfuric acid. The $D_{pore}$ of all the carbon precursors and catalysts were compared: they were almost the same at around 3.85 nm, indicating that the obtained catalysts were mesopore-predominant materials, and the carbonization temperature (under 350 °C) and sulfonation temperature (under 80 °C) had little

effect on the average pore diameter of the carbon bulk.

　　Figure 3 shows the pore size distribution and $N_2$ adsorption–desorption isotherms for all the carbon precursor and catalysts. All of the isotherms of the carbon precursor and catalysts were close to type IV ones according to the IUPAC (2015) classification[15]. The pore size distribution showed that the major pore diameter was around 3.85 nm for all the carbon precursor and catalysts, and there were almost no micropores. The adsorption–desorption hysteresis loops were indistinct due to the average pore diameters of catalysts being between the microporous size and the mesoporous size.

　　Combining the TG, SEM, and $N_2$-physisorption analysis, it can be concluded that the structure of RS degraded when the carbonization temperature was over 300 °C. The RS had weak mechanical strength due to the low content of lignin, owing to which the carbon precursor with abundant porous structure could be formed under low-temperature carbonization, and the overall structure did not collapse after sulfonation treatment, and even the $S_{BET}$ and $V_{Total}$ became greatly increased. The $S_{BET}$ and $V_{Total}$ of RS300-80 reached 420.9 $m^2/g$ and 4.05 $cm^3/g$, respectively. The high surface area and pore volume of the catalysts provided a stronger physical adsorption capacity, which improved the mass transfer by increasing the diffusion of the reactants. Meanwhile, as the carbonization temperature increased, aromatic carbon sheets were more uniform and rigid in space, which made it difficult to load $–SO_3H$ groups[16]. Therefore, we choose the precursor carbonized at 300 °C for the following experiments.

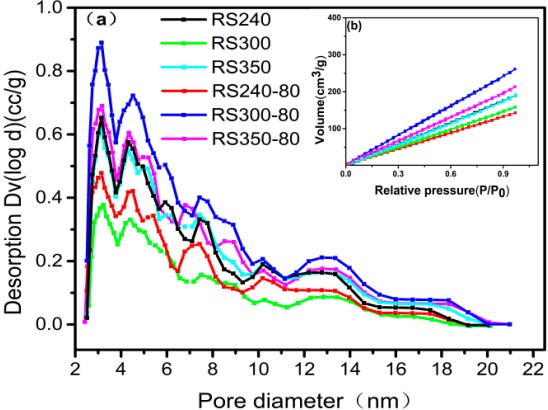

**Figure 3.** Pore size distributions (**a**) and $N_2$ adsorption-desorption isotherms (**b**) of all the carbon precursor and catalysts with different carbonization temperature.

**Table 1.** Properties of carbon precursors and prepared catalysts.

| Catalysts | $S_{BET}$ $(m^2/g)$ | $V_{Total}$ $(cm^3/g)$ | $D_{pore}$ (nm) |
|:---:|:---:|:---:|:---:|
| RS240 | 306.8 | 2.94 | 3.82 |
| RS300 | 260.5 | 2.44 | 3.74 |
| RS350 | 312.1 | 2.95 | 3.76 |
| RS240-80 | 222.1 | 2.22 | 3.98 |
| RS300-80 | 420.9 | 4.05 | 3.84 |
| RS350-80 | 337.7 | 3.30 | 3.90 |

## 2.4. FT-IR Analysis

FT-IR spectra of the RS300, RS300-50, RS300-80, and RS300-120 are shown in Figure 4. All the spectra exhibited the same peaks at around 1100 cm$^{-1}$ attributed to cyclic ethers (e.g., D-glucose, glycosidic bond) C–O–C bonds in the carbon precursor and catalysts. The C=C stretching vibration of the aromatic skeleton adsorption bands were at 1433, 1561, and 1697 cm$^{-1}$. The absorption bands at 1717 cm$^{-1}$ were attributed to the carbon–oxygen double bond stretching of the carboxylic acid group. The absorption bands observed at 3000–3600 cm$^{-1}$ were attributed to the alcoholic and phenolic –OH groups [17]. However, an obvious increase in the peak intensity was observed in the –OH groups spectra of catalysts. It may be that the formation of –OH groups was encouraged by the glycosidic bond disconnection of D-glucose of RS300 after sulfonation. Unlike the case involving RS300, the bands at 1037 cm$^{-1}$ (–SO$_2$ symmetrical stretching) and 1163 cm$^{-1}$ (–SO$_2$ asymmetrical stretching) could be observed for catalysts (e.g., RS300-50, RS300-80, and RS300-120), indicating the presence of a sulfonic group in the catalyst. These bonds confirmed the presence of the alcoholic and phenolic –OH groups, cyclic ethers C–O–C bonds, and aromatic skeleton in RS300, RS300-50, RS300-80, and RS300-120. In addition, these results also confirmed the incorporation of three different kinds of acids (e.g., –OH, –COOH, –SO$_3$H) in the carbon framework of the catalysts.

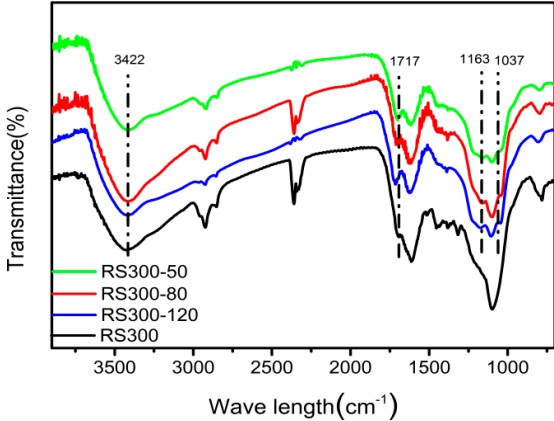

**Figure 4.** Fourier transform infrared spectroscopy (FT-IR) spectra of RS300, RS300-50, RS300-80, and RS300-120.

## 2.5. TPD Analysis

The NH$_3$-TPD tests of varying catalysts are shown in Figure 5. The acidities of weak, medium, and strong acids were estimated from the peak areas derived from a Gaussian fitting of the NH$_3$-TPD profiles in the temperature ranges of 50–190 °C, 190–400 °C, and >400 °C, respectively [18]. In general, the strong acid sites, medium acid sites, and weak acid sites are attributed to the presence of –SO$_3$H groups, –COOH groups, and –OH groups attached to the surface of catalysts [19]. FT-IR analysis also confirmed the similar result (Figure 4). The results of acidity value and weak, medium, and strong acid contents of different catalysts are summarized in Table 2. As can be seen, all the catalysts showed three obvious peaks in the ranges of 100 to 200 °C, 300 to 400 °C, and 500 to 600 °C, owing to the weak, medium, and strong acids, respectively. The total acidity and the strong acid site, and weak acid sites of catalysts desorption peaks gradually increased first and then decreased with an increasing temperature of sulfonation, while the formation of medium acid sites decreased. The results show that an appropriate increase in the sulfonation temperature is useful for increasing the strong acid sites, but the higher sulfonation temperature is counterproductive. During the sulfonation processes, the formation of strong acid sites may be due to the –OH groups of pyranose structure so that the cellulose remained through incomplete carbonization in RS300 and had strong electron-donating groups and the density of electronic cloud of surrounding carbon atom increased, leading to –SO$_3$H groups easily attaching to the carbon precursor through an electrophilic substitution reaction. However, there may be a reverse reaction of sulfonation at higher

temperature sulfonation, which is why only a few strong acid sites occurred at 120 °C. One possible explanation for a decrease in the density of –COOH groups was that an increased sulfonation temperature advanced the formation of anhydride between carboxylic acid derivatives by dehydration reaction in the concentrated $H_2SO_4$, which was not conducive to the formation of –COOH groups. However, as the sulfonation temperature increased (under 80 °C), the glycosidic bond in cellulose and hemicellulose of catalyst was sheared by concentrated acid hydrolysis, leading to the formation of weak acid sites, owing to which the weak acid sites increased. A further increase in the temperature could have led to a slight decrease in the density of –OH groups that advanced the esterification of carboxylic acid derivatives with hydroxyl groups at a high temperature. In contrast with other catalysts, the total acidity of RS300-80 reached 2.87 mmol/g. This is explained by the fact that the –SO₃H groups and –OH groups of RS300 gradually increased with increasing temperature of sulfonation (< 80 °C). Thus leading to the total acidity of catalysts increased. The –SO₃H groups, –COOH groups, and –OH groups of the RS300 decreases evidently after undergoing further increasing sulfonation temperature, hence the total acidity of catalysts decreased. This is probably due to the reverse reaction of the sulfonation reaction, the condensation reaction between –COOH groups, and the esterification reaction between –OH and –COOH groups at high sulfonation temperatures.

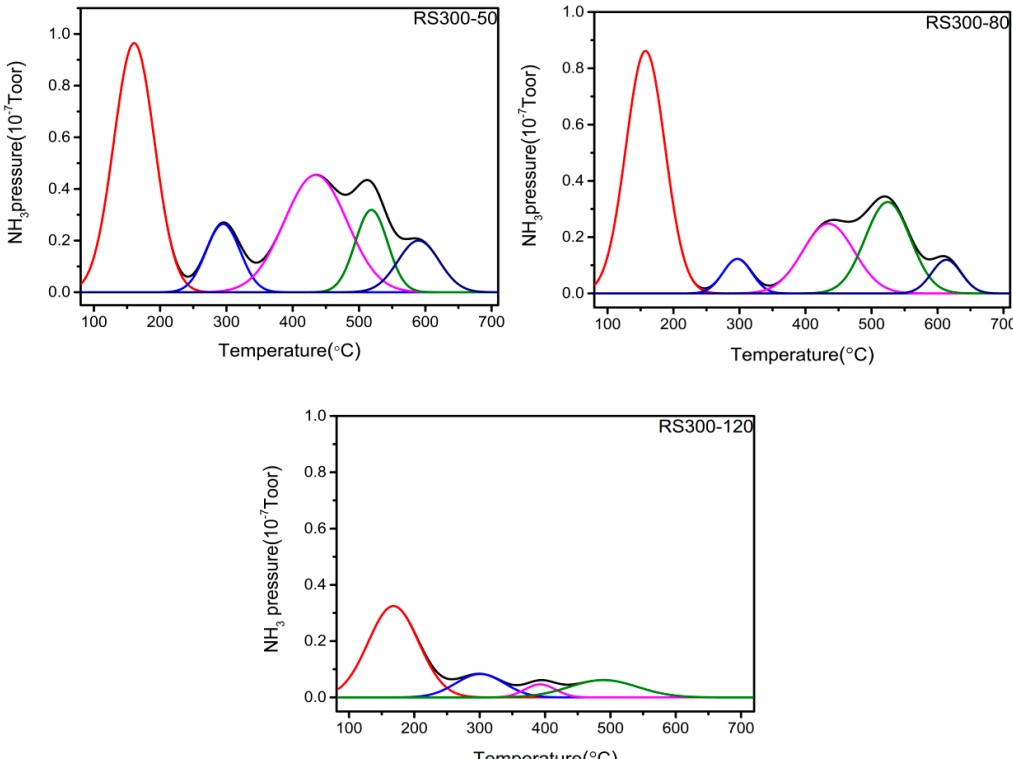

**Figure 5.** Temperature-programmed desorption (NH₃-TPD) patterns of RS300-50, RS300-80, and RS300-120.

**Table 2.** Acidity and strength distribution of catalysts.

| Catalyst | Total acidity/mmol/g | Percentage of acid sites (% of total acid) | | |
| --- | --- | --- | --- | --- |
| | | Weak acid (50–190 °C) | Medium acid (190–400 °C) | Strong acid (> 400 °C) |
| RS300-50 | 2.07 | 36.52 | 36.45 | 27.03 |
| RS300-80 | 2.87 | 48.25 | 23.13 | 28.62 |
| RS300-120 | 1.67 | 67.37 | 13.15 | 19.48 |

*2.6. XPS Analysis*

In Figure 6, XPS tests were conducted to characterize C 1 s, O 1 s, and S 2 p chemical states of RS300-50, RS300-80, and RS300-120. The peaks with a binding energy of 283.6, 285.5, and 286-287 eV were attributed to aromatic carbons (C=C), aliphatic carbons (–C–C/C–H), hemiacetal, and carboxylic carbon (–C=O), respectively [20]. The C=O, –OH, and –O–C type oxygen peaks were located at 531.7, 532.9, and 533.7 eV, respectively. The binding energy of S 2 p (e.g., 167.5 eV) shifted to a higher wavenumber compared with the electron-binding energy atlas (e.g., 164 eV), indicating that the sulfur element in the catalyst had a positive charge, and almost all of the S atoms in catalyst were contained in –SO$_3$H groups. It provided further evidence that the sulfonic acid group had been successfully loaded onto the catalyst. Hence, the densities of –SO$_3$H sites in the carbohydrate-derived catalysts could be calculated based on the S content in the catalyst [21].

The results indicated that RS300-50, RS300-80, and RS300-120 were composed of CS$_{0.029}$O$_{0.422}$, CS$_{0.04}$O$_{0.356}$, and CS$_{0.022}$O$_{0.359}$, respectively. Table 3 shows the quantitative results of chemical states of carbon, sulfur, and oxygen for the surfaces of RS300-50, RS300-80, and RS300-120. It showed that the C=C type carbon content was little affected by the increased sulfonation temperature, which means that sulfonation temperature causes almost no change in the aromaticity of the catalysts, and they have low aromaticity. The relationship between the surface S and –OH contents and the sulfonation temperature were consistent with TPD analysis. The contents of –O–C type oxygen of catalysts decreased first and then increased with increasing sulfonation temperature; however, the trend for –C–C/C–H type carbon was the opposite, indicating that RS300-80 may possess more alkyl side chains and less of the β-D-pyran structure [22]. It can be explained that in the extent of oxidation ring opening of aldehydes and primary alcohols in the D-glucose structure of cellulose of RS300 increases with the sulfonation temperature in concentrated sulfuric acid, so that RS300-80 contains more alkyl side chains and –OH groups, while –O–C groups decrease. Further increasing the sulfonation temperature, the complex reactions between –OH and –COOH groups include dehydration, esterification and the formation of anhydride, resulting in the alkyl side chain decreases and the –O–C groups increases.

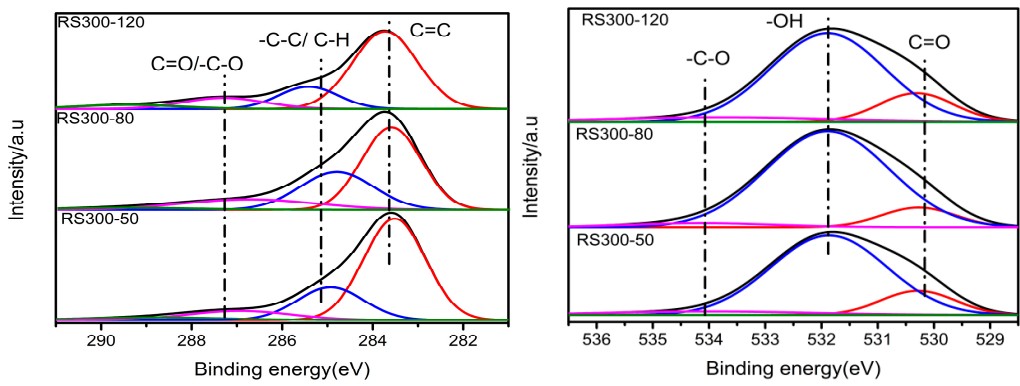

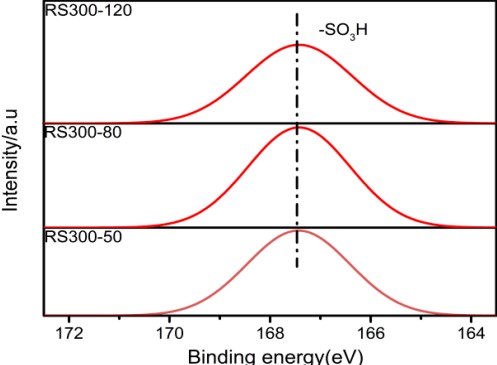

**Figure 6.** C l s, O 1 s, and S 2 p X-ray photoelectron spectroscopy (XPS) spectra obtained for the RS300-50, RS300-80, and RS300-120.

**Table 3.** Carbon, sulfur, and oxygen contents of RS300-50, RS300-80, and RS300-120 determined by XPS.

| .Catalyst | C (wt %) | | | O (wt %) | | | S (wt %) |
|---|---|---|---|---|---|---|---|
| | **C=C** | **–C–C/C–H** | **–C=O** | **C=O** | **–OH** | **–O–C** | **–SO₃H** |
| RS300-50 | 28.13 | 8.53 | 7.22 | 8.27 | 40.16 | 4.09 | 3.01 |
| RS300-80 | 27.31 | 12.33 | 8.74 | 5.13 | 41.11 | 2.20 | 3.16 |
| RS300-120 | 30.41 | 10.87 | 6.78 | 7.43 | 37.89 | 3.60 | 2.59 |

## 3. Materials and Methods

### 3.1. Material

$\alpha$-Pinene (purity of 98%) was of analytical grade purchased from Macklin Chemical Reagent Factory (Shanghai, China). Concentrated sulfuric acid (98%), acetone, and alcohol were of analytical grade purchased from Xi Long Scientific Co., Ltd., (Guangdong, China). The rice straws (RS) was from the experimental field of Guangxi University in Nanning, China. The RS were collected by manual harvesting in October 2018. The cross-section diameter and length of RS were about 3 cm and 45 cm, respective. Before vacuum storage, the RS was dried at 105 °C for 8 h.

### 3.2. Preparation of Catalyst

Biomass solid acid catalysts were prepared by the sulfonation of incomplete carbonized biomass. As a typical run, rice straw (RS) was cut into small sticks, grounded using a mixer grinder (<40 mesh), and then dried at 100 °C overnight in order to remove free water. The dried biomass (5 g) was immediately transferred to the quartz boat of a tube furnace, heated for 2 h, and then carbonized at different temperature with $N_2$ protection to produce brown-black carbon precursors solid denoted as RS*X* where '*X*' is the carbonization temperature. The carbon precursors were ultrasonically cleaned in alcohol to remove the emerging tar of incomplete carbonization and then dried for 5 h at 80 °C. The obtained samples were impregnated in the three round-bottomed flasks with $H_2SO_4$ (98% per gram powder join to 10 mL of concentrated $H_2SO_4$) and 300 rpm stirring at different temperature for 3 h. The sulfonated black solid laves were separated by vacuum filtration, washed with hot deionized water (>80 °C), and then collected in an evaporating dish. The catalysts (denoted as RS*X-Y*, where '*Y*' is the sulfonation temperature) were obtained by drying solid samples at 100 °C overnight.

### 3.3. Characterization of the Catalysts

The thermal stability of samples was measured by simultaneous thermo gravimetric (TG) analyzer under $N_2$ (30 mL/min) flow at a heating rate of 10 °C/min from 30 to 1000 °C (NETZSCH STA 449 F5, Serbo, Bavaria, Germany).

Scanning electron microscope (SEM) (Beijing Pricey Instruments Co., Ltd., Beijing, China) was used to characterize the morphological characteristics of samples. Samples were photographed after gold spraying using an ion sputtering apparatus.

The specific surface area and pore volume of samples were detected by $N_2$–physisorption (BET) (Quantachrome NOVA 2200e, Boynton Beach, Florida, USA) at −196 °C. Prior to the measurements, all the catalysts were degassed at 100 °C under a nitrogen atmosphere for 5 h. The Brunauer–Emmet–Teller (BET) equation and Barrett–Joyner–Halenda (BJH) model were used to calculate the specific surface area ($S_{BET}$), average pore diameter ($D_{pore}$), and total pore volume ($V_{Total}$), respectively.

Ammonia temperature-programmed desorption ($NH_3$-TPD) experiments were carried out using a U-type quartz tube reactor equipped with a residual gas detector (RGA, RGA200, Agilent, Palo Alto, California, USA). Before the adsorption experiments, the samples (0.1 g) were purged at 200 °C for 1 h in an $N_2$ flow. Upon cooling to 80 °C, the prepared samples were saturated with an $NH_3$ flow for 1 h, and the physisorbed $NH_3$ was removed by purging with $N_2$ gas for 1 h. The $NH_3$ desorption was performed by heating the sample from 50 to 800 °C at a heating rate of 10 °C/min in $N_2$ flow. The desorbed $NH_3$ was analyzed on-line with the RGA.

Fourier transform infrared spectroscopy (FT-IR) of catalysts were performed by Thermo Trace 1310-Nicolet IS50 spectrometer (Nicolet 6700, Thermo Fisher Scientific, Shanghai, China). The FT-IR resolution and number of scans were set as 4 cm$^{-1}$ and 40 scans per spectrum, respectively. The FT-IR spectra were recorded in the range of 4000–500 cm$^{-1}$. The samples were grounded with KBr power and pressed into pellets to conduct the FT-IR test.

The carbon, oxygen, and sulfur element of samples were determined using X-ray photoelectron spectroscopy (XPS) under vacuum. XPS was equipped with a monochromatic Al X-ray source operating at 6 μA × 3 keV and with the signal averaged over an oval-shaped area of approximately 700 × 300 microns (ESCALAB 250XI+, Thermo Fisher Scientific, Shanghai, China). The recorded photoelectron binding energies were referenced against the C 1s contamination line at 284 eV.

### 3.4. Investigatment of Catalytic Activity

The hydration experiments of $\alpha$-pinene were carried out in a three-necked flask at 80 °C for 24 h. As a typical run, 5 mL of $\alpha$-pinene, 5.25 mL of deionized water, 0.5 g of catalysts, and 20 mL of solvent were added to reaction vessel with 300 rpm stirring. The solid sample was recycled by a centrifuge.

### 3.5. Products Analysis

The liquid products were analyzed by gas chromatography (Shimadzu GC-2010, Japan), which was equipped with a flame ionization detector. Separations were carried out on an acidic functional groups and polymer skeleton of polar capillary column GC-2010 (30 m × 0.25 mm × 0.25 μm; Shimadzu, Japan). The temperature of the oven was set at 70 °C, increased to 180 °C with 2 °C/min, and held for 5 min; then, it was increased at 5 °C/min up to 220 °C and held for 5 min. The split ratio was 1:50 and the carrier gas was nitrogen. The injector and detector temperatures were 240 °C and 280 °C, respectively. The reaction mixture was centrifuged to remove the solid catalyst and the oil phase. Then, a 0.2 μL sample of oil phase was injected into GC. The peaks of chromatogram were identified by comparing retention times and gas chromatography-mass spectrometry. The selectivity of $\alpha$-terpineol and conversion of $\alpha$-pinene were calculated by the external standard method, the standard curves was prepared by detecting different concentration of standard solution, and the external standards was carvone. The conversion of $\alpha$-pinene (X) and the selectivity (S) of $\alpha$-terpineol were calculated using the following formula:

$$X \;=\; \frac{m_{pinene}}{m^{0}_{pinene}} \times 100 \;\; \% \tag{1}$$

$$S \;=\; \frac{Y_{\alpha\text{-}terpilenol}}{X} \times 100 \;\; \% \tag{2}$$

where $m^{0}_{\alpha\text{-}pinene}$ was the initially added volume of the $\alpha$-pinene, which is converted into the mass of $\alpha$-pinene (g), $m_{\alpha\text{-}pinene}$ was the mass of $\alpha$-pinene consumed (g), and $Y_{\alpha\text{-}terpineol}$ was yield of $\alpha$-terpineol (%).

## 4. Activity Tests of Catalysts

### 4.1. Effect of Carbonization Temperature

To study the effect of carbonization temperature on the catalyst activity, RS was carbonized at different temperatures in the range of 240 to 350 °C. The catalytic performance of carbon-based catalysts was evaluated at 80 °C for 24 h, and the results are listed in Table 4. The other parameters (0.5 g RS300-80, 10:1 water/$\alpha$-pinene molar ratio) were fixed, and the desired product was $\alpha$-terpineol. The results showed that the conversion of $\alpha$-pinene increased with the increasing carbonization temperature. However, the selectivity of $\alpha$-terpineol first increased and then decreased. As can be observed in Table 4, at the temperature of 300 °C, the selectivity of $\alpha$-terpineol reached the highest value of 57.07%. This phenomenon could be due to the optimal active sites and good pore structure formed at 300 °C. When the carbonization temperature was lower than 300 °C, the lower catalytic activity caused the absent of tubular porous structures in the interior of RS240-80. Thus, the reaction of hydration of $\alpha$-pinene only occurred on the surface of RS240-80. When the carbonization temperature was higher than 300 °C, although the conversion of $\alpha$-pinene reached a maximum of 84.16%, the selectivity of $\alpha$-terpineol was only 13.20%. This may be explained by the higher aromaticity of RS at high-temperature carbonization, –SO$_3$H group as the main active compound were too difficult to be loaded onto the structure of catalysts. The low surface area and pore volume of the RS350-80 also hindered the separation of $\alpha$-terpineol from active site and promoted the formation of byproducts from the hydrogenation of $\alpha$-terpineol [23].

### 4.2. Effect of Sulfonation Temperature

To observe the effect of the sulfonation temperature on the catalyst activity, catalysts were prepared at different sulfonation temperatures: 50 °C, 80 °C, and 120 °C, and the desired product was $\alpha$-terpineol (Table 4) The other parameters (0.5 g RS300-80, 10:1 water/$\alpha$-pinene molar ratio, 80 °C, 24 h) were fixed. With increasing sulfonation temperature, the conversion of $\alpha$-pinene increased and the selectivity of $\alpha$-terpineol first increased and then decreased. The result showed that RS300-80 had a selectivity of $\alpha$-terpineol up to 57.07% and the conversion of $\alpha$-pinene reached 67.60%. As indicated by the NH$_3$-TPD analysis, the contents of strong and weak acids gradually increased and then decreased in RS300 with the increasing temperature of sulfonation, and it reached a maximum of 80 °C. Therefore, it can be deduced that the selectivity of $\alpha$-terpineol and the conversion of $\alpha$-pinene increased with increasing strong and weak acid contents, respectively. It can be surmised that the catalytic sites of all the catalysts are mainly weak acid sites, and most of the $\alpha$-pinene could collide easily with the weak acid molecules and then generate the carbon positive ion intermediate, leading to the conversion rate of $\alpha$-pinene increasing with an increase in the content of weak acid sites. According to the reaction mechanism diagram analysis, the formation of hydrated terylene glycol by an addition reaction between the inner double bond of $\alpha$-terpineol with water further increased the content of $\beta$-terpineol and $\gamma$-terpineol. Hence, the selectivity of $\alpha$-terpineol deceased. However, the strong acid site can effectively inhibit the occurrence of the reaction, which is attributed to the dehydration of sulfonic acid group, resulting in increased selectivity of $\alpha$-terpineol.

Hence, when the sulfonation temperature was below 80 °C, RS300-50 offered a lower conversion and selectivity of $\alpha$-terpineol, which was possibly because it could not provide enough strong and weak acid activity sites, leading to a lower number of active sites for $\alpha$-pinene conversion. Although RS300-120 exhibited a higher conversion of $\alpha$-pinene with a maximum of 74.05%, the selectivity of $\alpha$-terpineol was 34.02%. This may be due to sulfonation of polycyclic aromatic carbon was reversible exothermic electrophilic substitution reaction [24], the high sulfonation temperature produced unstable and hydrolyzable sulfonation products, and a reversible reaction occurred with sulfonic acid group breaking away from carbon bulk. Thus leading to the low content of $-SO_3H$ group in RS300-120, but it had enough weak acid sites. Thus, more by-products were formed, thus increasing the conversion of $\alpha$-pinene.

In summary, RS300-80 had a high acid site density, high porosity, and large pore size, which was considered a high catalytic ability for the synthesis of $\alpha$-terpineol. Thus, RS300-80 was used for the subsequent experiments.

**Table 4.** Performance of carbon-base solid acid catalysts in the hydration of $\alpha$-pinene.

| Catalyst | Conversion (%) | Selectivity (%) |
|---|---|---|
| RS240-80 | 57.23 | 35.27 |
| RS300-80 | 67.60 | 57.07 |
| RS350-80 | 84.16 | 13.20 |
| RS300-50 | 55.33 | 36.85 |
| RS300-120 | 74.05 | 34.02 |

### 4.3. Effect of Reaction Temperature

The effect of reaction temperature on the conversion and selectivity of $\alpha$-terpineol over RS300-80 was investigated. The reaction was performed under fixed parameters: 0.5 g RS300-80, 10:1 water/$\alpha$-pinene molar ratio, 24 h, and 20 mL solvent, and the results are presented in Figure 7a. As the reaction temperature increased up to 80 °C, the selectivity of $\alpha$-terpineol reached a maximum of 57.07% and marginally affected the conversion of $\alpha$-pinene. Further increment of temperature above 80 °C showed a decrease in the selectivity of $\alpha$-terpineol and the conversion of $\alpha$-pinene increased. The $\alpha$-pinene carbocation intermediate underwent a ring opening reaction of its bicyclic structure to form another carbocation intermediate, which requires a higher energy, hence the selectivity of $\alpha$-terpineol lower when temperature below 80 °C. An excessively high reaction temperature would also intensify the evaporation rate of acetone into the gas phase, resulting in the unavailability of the required amount of intermediate products for sufficient contact with water, hence reducing the selectivity. The yield of $\alpha$-terpineol decreased when the reaction temperature became too high because the hydration reaction of $\alpha$-pinene is a reversible exothermic reaction [25]. Besides, the conversion of $\alpha$-pinene increased, which may be the result of positive mass transfer between $\alpha$-pinene and the active sites of RS300-80. Ascribe to high temperature intensifies the movement of $\alpha$-pinene molecules, thus leading to promote the hydration reaction rate of $\alpha$-pinene. Overall, the optimum reaction temperature for the hydration of $\alpha$-pinene in the presence of biomass catalyst was found to be 80 °C in this study.

### 4.4. Effect of Reaction Time

The reaction time affects the selectivity of $\alpha$-terpineol and cost of production. The catalytic performance of RS300-80 was investigated for reaction times of 20 to 28 h. The study was conducted using water/$\alpha$-pinene molar ratio of 10:1, 0.5 g RS300-50, and 80 °C. The selectivity of $\alpha$-terpineol and conversion of $\alpha$-pinene are shown in Figure 7b. In Figure 7b, the conversion of $\alpha$-pinene and the selectivity of $\alpha$-terpineol can be seen to have obviously increased from 20 to 24 h. The maximum

selectivity of α-terpineol was 57.07% at 24 h, and the conversion of α-pinene was 67.60%. The conversion of α-pinene increased from 67.60% to a maximum of 90.08% at 28 h. However, any further increment in reaction time resulted in a rapid decrease in the selectivity of α-terpineol instead. This may be explained by the transformation of α-terpineol into other products through the hydration reaction, leading to more by-product formation, which is a similar conclusion to the research results of references [26]. According to the reaction pathways (Figure 8) and Table 5, they can be inferred from the side that when the content of α-terpineol decreases, the content of by-products does increase, especially 1,3-cyclohexadiene, D-limonene, and terpinolene. Table 5 data was determined by GC-MS. GC-MS can achieve qualitative results, and the relative content given by GC-MS can approximate quantitative results. The result showed that the optimum reaction time was 24 h.

### 4.5. Effect of Water/α-Pinene Molar Ratio

The effect of water/α-pinene molar ratio plays an important role in the selectivity of α-terpineol. Effect of water/α-pinene molar ratio on the hydration of α-pinene was studied by applying different molar ratios (5:1, 10:1 and 15:1) to a fixed process parameter (0.5 g RS300-80, 80 °C, 24h), as shown in Figure 7c. When the water/α-pinene molar ratio increased from 5:1 to 15:1, the conversion of α-pinene changed little, which showed that water/α-pinene molar ratio had little effect on the conversion of α-pinene. But the selectivity of α-terpineol first increased and then decreased. When the water/α-pinene molar ratio was increased up to 10:1, the selectivity increased to a maximum of 57.07%. This can explain that the hydration reaction of α-pinene preferentially forms carbocations formed by the hydrogen ions of RS300-80 attacking the α-pinene molecules. The number of carbocations formed determines the conversion of α-pinene, and it can be inferred that the conversion of α-pinene was related to the acidic site of RS300-80, rather than water molecule. Therefore, increasing the content of water molecules has little effect on the α-pinene conversion. However, the increase in water molecule content increases the contact surface between carbocations and water, resulting in an increase in the selectivity of α-terpineol. When the water/α-pinene molar ratio was increased up to 15:1, the selectivity of α-terpineol decreased. This may be explained that water could promote the generation of α-terpineol, but the excess water might increase the formation of byproducts (e.g., borneol and fenchyl alcohol).

### 4.6. Effect of Catalyst Loading

Since the presence of a catalyst reduces the activation energy of reaction, it is necessary to investigate the effect of different loading amounts of RS300-80 on the hydration of α-pinene. The reaction was performed under fixed parameters: 10:1 water/α-pinene molar ratio, for 80 °C at 24 h. The experiments were conducted with different RS300-80 dosages (0.25, 0.50, and 0.75 g). The effect of RS300-80 dosage on the catalytic performance is shown in Figure 7d. The rate of α-pinene conversion and the selectivity of α-terpineol followed a linear relationship with the dosage of RS300-80 when the RS300-80 dosage was under 0.5 g. This could be attributed to the presence of more acidic active sites with increasing RS300-80 dosage. When the dosage of RS300-80 was increased from 0.5 to 0.75 g, the conversion value remained increased, but the α-terpineol yield evidently decreased. This discrepancy may be caused by the increase of acidic catalytic sites in the reaction system with the increase of the loading of RS300-80 led to the α-pinene molecules being encouraged to generate more carbocations. As a result, conversion rates continue to increase. However, higher amount of RS300-80 loading would lead to negative mass transfer between ring opening carbocations and water molecules[27]. Causes the selectivity of α-terpineol to decreased.

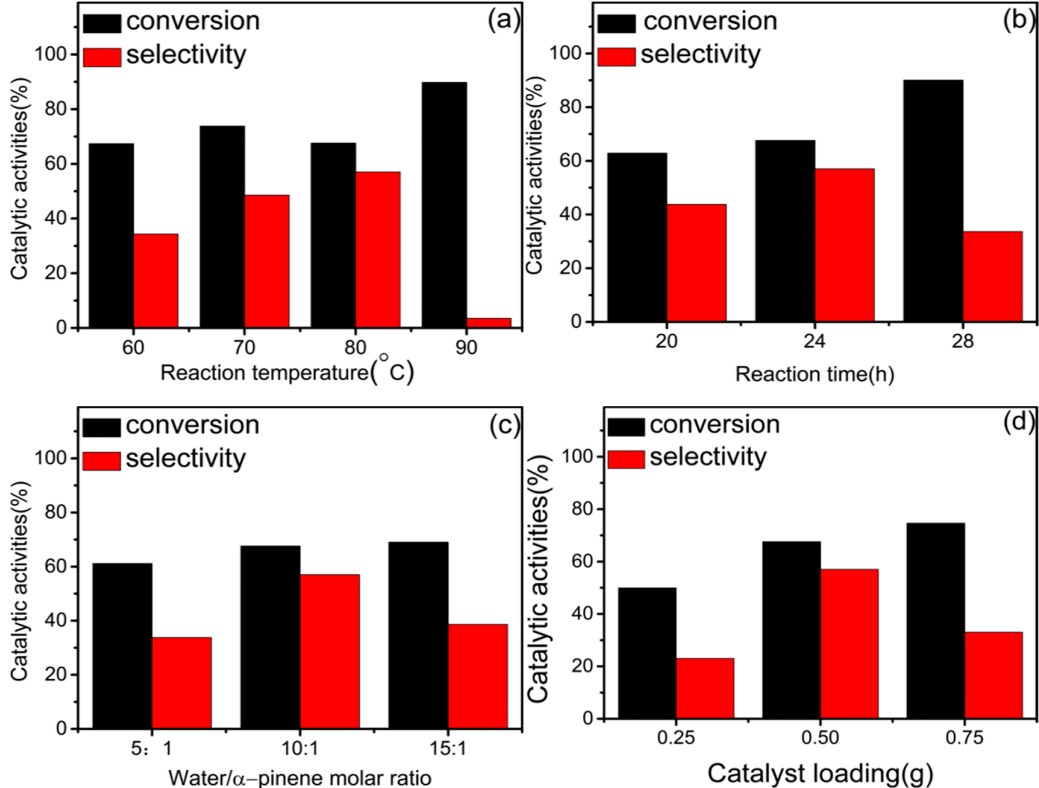

**Figure 7.** Performance of biomass carbon-based solid acid catalyst: (**a**) effect of reaction temperature; (**b**) effect of reaction time; (**c**) effect of water/α-pinene molar ratio; and (**d**) effect of catalyst loading. Other reaction conditions were held constant.

## 5. Postulated Reaction Mechanism

The compositions of mixed liquid products obtained in the hydration of α-pinene at 80 °C in the solvent and in the absence of solvent are shown in Table 5. The reaction was performed under fixed parameters: 0.5 g RS300-80, 10:1 water/α-pinene molar ratio, and 24 h. Compared with the presence of solvent-containing of liquid products, the solvent-free liquid products has a higher by-product content. The results revealed that the presence of solvent could increase the content of α-terpineol and decrease the formation of by-products. This may be because the content of α-terpineol was increased by an addition reaction that took place between the six-membered ring branched-chain double bond of by-products (e.g., D-Limonene, terpinolene) and water. The hydrogenation reaction leads to the formation of a large of carbonium ion of α-pinene when the α-pinene molecule enters RS300-80, which is attributed to the catalyst having a large amount of catalytic groups (e.g., –OH, –COOH, and –SO₃H). Besides, although –OH, –COOH, and –SO₃H are hydrophilic groups [28], they allowed a large number of water molecule into RS300-80, but the content of α-terpineol still was low in such cases in the absence of solvent. This is due to the hydrophobic nature of the α-pinene carbocation, which results in insufficient contact between it and water molecules in the absence of a solvent. Therefore, the content of α-terpineol decreased. Figure 8 shows that the products were mainly six-membered C10 hydrocarbon rings such as α-terpineol and its derivatives. The reduced yield of by-products confirmed the occurrence of the above-mentioned reactions. The conversion of α-pinene without solvent was higher than that with solvent. This can be interpreted as the formation by-products being encouraged in the absence of a solvent. The carbonium ion with a positive charge easily attracted the RS300-80 because the oxygen of the β-D-pyran of RS300-80 has a higher electronegativity. Besides, the carbonium ion of α-pinene could not fully connect with water. It tended to produce more positive carbon ions as products, which led to the contents of α-pinene decreasing. Overall, α-pinene underwent the

formation of carbon cations of at least five different types. Two of them are without the open loop of carbon cations, and they yielded fenchyl alcohol and borneol, respectively. The others yielded α-terpineol or underwent the dehydrogenation to yield cinene directly, which would form α-terpinene in the presence of the hydrogen ion of catalysts. The last type was D-limonene formed by the dehydrogenation reaction.

As indicated by XPS and FT-IR analysis, the presence of cyclic ether (e.g., D-glucose, glycosidic bonds) C–O–C bonds in RS300-80 indicates that the process of carbonation (at 300 °C) did not completely damage the β-D-pyran of cellulose and hemicellulose. The structure of phenyl rings of lignin did not undergo excessive disintegration at low-temperature carbonization. This led to the formation of an amorphous carbon structure composed of aromatic carbon sheets and pyran, and the –SO₃H, –COOH, and phenolic –OH groups were loaded on it. The postulated molecular structure of RS300-80 is shown in Figure 9.

**Table 5.** Effect of solvent on composition and product distribution of catalytic hydration of α-pinene: (**a**) without solvent; (**b**) with solvent.

| NO. | Product | Molecular formula | Relative content (%) | | Similarity (%) |
|---|---|---|---|---|---|
| | | | (a) | (b) | |
| 1 | α-Pinene | $C_{10}H_{16}$ | 7.23 | 17.33 | 97 |
| 2 | Camphene | $C_{10}H_{16}$ | 9.80 | 3.69 | 97 |
| 3 | 1,3-Cyclohexadiene | $C_{10}H_{16}$ | 7.23 | 0.53 | 96 |
| 4 | D-Limonene | $C_{10}H_{16}$ | 15.95 | 8.30 | 90 |
| 5 | Eucalyptol | $C_{10}H_{18}O$ | 1.69 | 1.63 | 93 |
| 6 | 1,4-Cyclohexadiene | $C_{10}H_{16}$ | 3.98 | 3.97 | 97 |
| 7 | Terpinolene | $C_{10}H_{16}$ | 14.10 | 9.19 | 96 |
| 8 | 2-Norbornanol | $C_{10}H_{18}O$ | 8.14 | 6.97 | 96 |
| 9 | β-Terpineol | $C_{10}H_{18}O$ | 2.37 | 2.20 | 93 |
| 10 | Isoborneol | $C_{10}H_{18}O$ | 13.34 | 1.55 | 97 |
| 11 | α-Terpineol | $C_{10}H_{18}O$ | 16.17 | 44.64 | 86 |

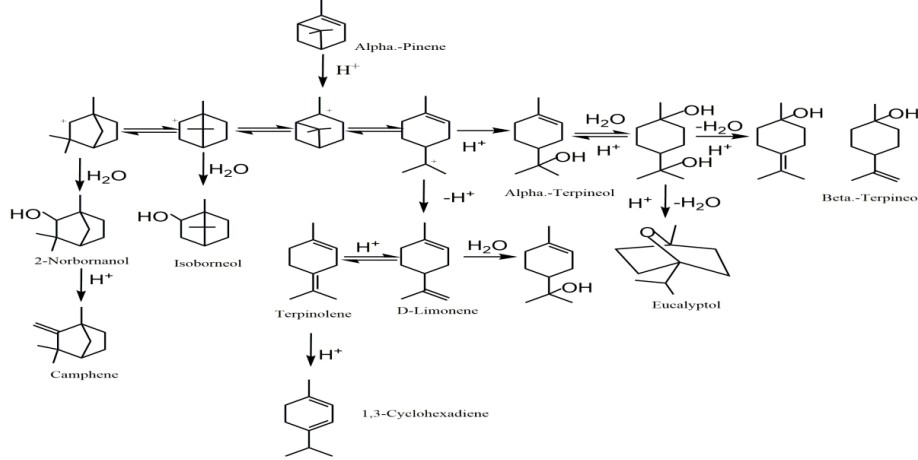

**Figure 8.** Reaction pathways involved in the conversion of α-pinene into hydrocarbons.

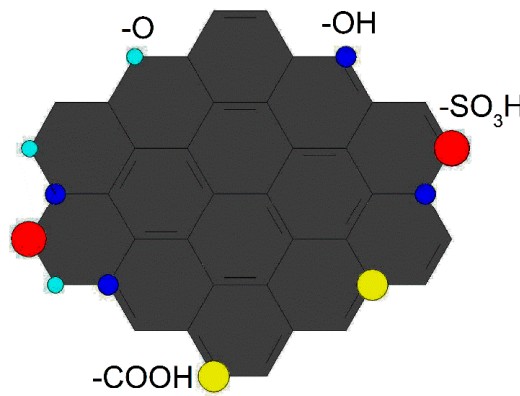

**Figure 9.** Structure of carbonized rice straw (RS) after sulfonation.

## 6. Conclusions

A carbon-based solid acid catalyst was prepared by rice straw; it was successfully used to conduct the hydration of $\alpha$-pinene. At optimum reaction parameters of 0.5 g catalyst loading, a 10:1 water/$\alpha$-pinene molar ratio, 24 h reaction time, and 80 °C reaction temperature, the selectivity of $\alpha$-terpineol and the conversion of $\alpha$-pinene reached 57.07% and 67.60%, respectively. The temperature of the carbonization and sulfonation treatment have a profound influence on the morphology, acid property, and catalytic activity of the obtained catalysts. The carbon-based solid acid catalysts with the highest strong acid density, high surface area, and pore volume (RS300-80) exhibit the highest selectivity of $\alpha$-terpineol. The RS300-80 had abundant strong acid sites (0.82 mmol/g), and the $S_{BET}$ and $V_{Total}$ of catalysts reached 420.9 m²/g and 4.05 cm³/g which were conducive to the high catalytic ability of the catalyst. Waste straw was the raw material for catalyst, and the preparation of catalysts required a lower energy consumption, in which the carbonization and sulfonation temperature were 300 °C and 80 °C, respectively. This shows that this method has the advantages of high cost-effectiveness, high safety, easy operation, and it is environmentally friendly. Meanwhile, the agricultural waste biomass was avoided to be used as a low-value energy resource. Some kinds of reaction routes for $\alpha$-pinene conversion such as the addition reaction, hydration, and dehydrogenation were discussed based on the experimental data.

**Author Contributions:** Z.W. designed, performed the experiments, analyzed the data, and wrote the paper; D.X conceived the experiments; P.D. and L.L. carried out SEM, BET, TG, TPD, XPS, and FT-IR analysis and helped with the interpretation; S.D. and Y.L. revised the manuscript; P.N. and X.C. assist software to process the data. All authors have read and agreed to the published version of the manuscript.

**Funding:** This research was funded by the Guangxi Nature Science Foundation of China (the project number 2017Z003).

**Conflicts of Interest:** The authors declare no conflict of interest.

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
