# Peer review of "Preparation of Carbon-Based Solid Acid Catalysts Using Rice Straw Biomass and Their Application in Hydration of α-Pinene"

_catalysts, doi:10.3390/catal10020213_

Round 1

Reviewer 1 Report

January 2, 2020

This manuscript presents the effect of carbon-based solid acid for the conversion of α-pinene using rice straw and their properties were investigated. This study is somehow interesting for readers of Catalysts journal; however, it is not enough to be accepted to the journal. The reviewer has the multiple queries listed below to improve the manuscript that should be replied in an exhaustive and sound basis before considering the article for publication.

1) There are great extensive data and publishable figure and table; however, they need to be professionally modified/changed to fit into the journal guidelines. The reviewer can see many major/minor errors in figure, table, typo, non-scientific words (good or bad), grammatical error, sentence, structure and others in the whole manuscript that should be corrected. In addition, the reference numbers and references are not matched, and some references are not completed.

2) Materials and methods parts need to be specific and provide more detailed information for researcher who wants to follow similar research. For example, rice straw should include year, month, location, particle size, moisture content, how the authors collected samples and stored after collected.

3) Analytic methods should be separated and described more specific with related references.

4) The authors described the data obtained from each test and analytical method; however, further research and comparison data are required in result section. The authors described several references in the manuscript; however, a high-level description with figures and table will improve the current study.

5) The reviewer wonders about the novelty of this study, why rice straw and carbon-based solid acid method are important for hydration of α-pinene compared to other methods and alternative agricultural biomass. To emphasize the current work, the authors need to add and present the key message and novelty of this study.

Reviewer 2 Report

This paper deals with the synthesis of carbon-based solid acid catalysts using rice straw and the hydration reaction of α-pinene using the synthesized catalysts. The authors elucidated the effects of carbonization temperature and post treatment temperature by sulfuric acid on the properties of catalysts. They also investigated the dependence of reaction conditions on the α-pinene conversion and the α-terpineol selectivity. The methods for both the characterization of catalysts and the reaction experiments are correct. However, several discussion parts contain a jump in logic and are not based on their experimental results. Therefore, I do not recommend this paper for publication in its present form.

P6, L200

The authors mentioned that well-developed pore structure of RS300-80 attributed the reaction because a large number of reactants could enter inside RS300-80. However, this discussion should be based on the relationship between the pore size of catalysts and the diffusion process in pores. Are the pore sizes of other two catalysts too small for reactants to diffuse?

P8, Table 1.

Is Dtotal mean average pore size? If so, not average value but pore size distribution is preferable for discussing pore structure.

P8-10, Section 3.4

From the TPD results, there are four or five peaks. Why the authors concluded there are three (not four or five) types of acid sites?

P14, L357

Why the selectivity of α-terpineol increased by the existence of strong acid sites and the conversion of α-pinene increased by the existence of weak acid sites? Did the α-pinene reaction to α-terpineol and that to α-pinene proceed different active sites?

P15, Section 4.4 and 4.5

The authors explained that hydration of α-terpineol to byproduct is the reason for decrease in α-terpineol selectivity. Did the authors confirmed by analyzing these byproducts?

Abstract, Introduction, and Conclusion section

It seems that many processes including calcination in high temperature and treating by sulfuric acid exist when synthesizing the catalysts. Please show the quantitative reasons why the authors considered this method as “high cost-effective, high safety, easy operation and environmentally friendly”?

There are some mistakes or insufficient description as follows. Please check the whole manuscript. Also, English in the manuscript should be improved.

P2, L46. “Maria et al.” cannot be found in the reference section.

P3, L97. What kind of “alcohol” did the authors use?

P4, L162. “Fiureg1” (misspelling)

P7, Caption of Figure 2. “RS540” (misspelling of RS240?)

Round 2

Reviewer 1 Report

The revised version of manuscript has been improved and addressed the reviewer's comments and suggestion. The reviewer believes that it is acceptable to Catalysts journal. There are a couple of suggestions the reviewer has that:

i) The authors need to check the whole manuscript again for correcting minor errors such as units, paragraphs and typos. The reviewer can still find some errors.

ii) Some more details are necessary in each legend of figure and table. The authors simply described what is the data in figure or table; however, it is important to explain/describe the experimental conditions and key comparisons in the legends for readers, even though a detailed explanation is in materials and methods or results sections. 

Reviewer 2 Report

The authors have satisfactorily addressed my questions and comments.  I recommend this article for acceptance to publish in Catalysts.

Author Response

Manuscript Title:Preparation of carbon-based solid acid catalysts with Rice Straw biomass and their application for hydration of α-pinene

Catalysts-687856

Dear Reviewers: Thank you for your letter and for the reviewers’ comments. We appreciate for Reviewers’ warm work earnestly, and professional guidance. Those professional guidance are valuable and very helpful for revising and improving our manuscript, as well as significance to our researches. Once again, thank you very much for your serious and professional work.

Thank you and best regards!

Yours sincerely

Prof. Deyuang Xiong

Zhao-zhou Wei